# Community Resilience Processes in Schools with Roma Students during COVID-19: Two Case Studies in Spain

Virginia Martínez-Lozano [1], Beatriz Macías-Gómez-Estern [1,*] and José L. Lalueza [2]

1 Faculty of Social Sciences, Universidad Pablo de Olavide, Ctra. Utrera km.1, 41013 Sevilla, Spain
2 Faculty of Psychology, Universidad Autónoma de Barcelona, Plaça Cívica, Bellaterra, 08193 Barcelona, Spain
* Correspondence: bmacgom@upo.es

**Abstract:** The aim of this paper is to describe coping strategies for difficulties generated by the COVID-19 pandemic in schools categorized as "highly vulnerable" in Spain, mainly focusing on children from immigrant and Roma families. Within the framework of a broader research study, we focus our gaze, as a case study, on two schools using in-depth interviews with its principals. These two schools were chosen as case studies because they had shown a history of carrying out documented educational change processes for at least 3 years before the pandemic. Our analytical strategy was a qualitative content analysis of a deductive–inductive nature. The results of our study can help identify key elements for the mechanisms of "Community Socio-Educational Resilience" and show that the innovative educational strategies tested by schools in the periods prior to the pandemic facilitated the generation of specific strategies for addressing problems arising from the pandemic. These strategies contributed to cushioning the increased inequality generated by the pandemic, contributing to the social sustainability of the educational and community system.

**Keywords:** resilience; vulnerable schools; COVID-19; culturally sensitive education; community-engaged research

## 1. Introduction

### 1.1. School Segregation and the Pandemic

One of the main obstacles to equity in the educational system is the existence of a group of schools that, due to the social composition of their students, represent a form of segregation, generally with a racialized basis. In the case of Spain, these schools are characterized by having a majority of the student body being from families from immigrant backgrounds or native Roma families [1,2]. The stigma that generally accompanies these schools causes the "flight" of non-Roma native families, which generates more segregation and more stigma for schools that host mostly (or exclusively) socially vulnerable populations. The isolation of vulnerable populations is particularly evident in the educational sphere, as early school failure or dropout mainly affects this population, which is mostly concentrated in segregated centers [3,4].

These vulnerabilities increase during social crises. Thus, during the first phase of the COVID-19 pandemic, characterized by the lockdown and closure of schools, a widespread perception was confirmed: The negative effects of this crisis increased the already-existing educational inequalities [5,6]. The studies developed during the lockdown showed that this crisis particularly impacted the most vulnerable sectors, widening the digital learning gaps that already existed before it and causing new ones [7,8]. It has been found that the income level of families determined their access to technology and was related to the hours devoted to homework, just as the educational level of parents was also related to the strategies available to them in supporting homework [5], and also that children and adolescents spent their time at home doing different activities depending on the family's instructional capital [9]. Other studies show the consequences of school closures on student learning,

and that these consequences were aggravated depending on the economic and employment situation at home or the level of parental stress, circumstances that introduce medium- and long-term effects on inequality and increase poverty [7,10]. In Spain, these difficulties affected especially public schools, where the number of students from disadvantaged socioeconomic contexts and a lack of technological resources is twice as high as in other types of centers [11].

### 1.2. Pedagogical Innovation and Community Socio-Educational Resilience

Understanding the problem of vulnerable minorities from an exclusively deficit approach can lead to a stigmatizing view: a perspective based on what the literature has described as the deficit perspective in education [12]. According to such a view, students from minority groups are systematically related to certain deficits, problems and shortcomings; they are affected by inadequate family socialization processes and present cultural limitations together with economic as well as linguistic deficiencies that all ultimately hinder their learning processes. To counter these deficit-based stances, and in reaction to the delay in the enforcement of effective policies that could end the segregation of vulnerable populations, some educational teams have launched educational change projects aimed at addressing inequality and promoting inclusion. The research question addressed in this study was whether these educational change processes that were implemented in pre-pandemic periods have contributed in any way to recovering the educational system post-COVID-19 crisis.

The pandemic and lockdown led to high levels of teacher stress and clear risks of collapse at personal and systemic levels [13]. Nevertheless, such situations can and should contribute to the creation of new learning conditions for different teaching systems [14]. Some studies were able to verify this outcome in the field of higher education [15,16], showing that the pandemic crisis, under certain conditions, may have served as an accelerator of innovation. This topic has been widely studied in relation to digitization. For example, a recent study demonstrated that the COVID-19 crisis expanded educators' use of digital communication tools, improving their teaching strategies after the pandemic: "It is a great opportunity to share knowledge and to perceive how technological tools can enable access to learning and complement presential education in the post-COVID-19 period" [17].

Thus, crisis situations such as the one produced by the pandemic can lead to the activation of functions that allow a system to recover from a serious disturbance. This leads us to the notion of community resilience, which refers to a community's ability to adapt and recover from difficult or traumatic situations [18]. This community resilience concept refers to two key elements that must be considered in our study. The first is the adoption of a social justice perspective, since inequalities and injustice contribute to the vulnerability of communities in crisis situations. Community resilience is based on addressing these underlying problems by encouraging the active and meaningful participation of communities in decision-making, and implementing measures that strengthen the capacity of communities to address the root causes of this vulnerability [19]. The second element of community resilience is the central role of the social network and the relevance of the connection between different systems and environments in which a person lives and develops. Reductionist conceptions of resilience as an individual characteristic that overlook social and community conditions are thus avoided.

Recently, the notion of "Community Socio-Educational Resilience" (CSER) [20] has been formulated specifically within the framework of research on the capacity of educational systems to address the pandemic. CSER has been defined as "the involvement of different social, cultural and educational agents in the design and implementation of creative-transformative educational responses to situations of adversity and uncertainty" (p. 2). This formulation, in line with the previous concept, situates resilience within its geographical locality and institutional networks. This is what Bronfenbrenner [21] calls a mesosystem: in the school environment, a mesosystem would correspond to the interactions between the school and families, as well as between families themselves. In fact,

Bronfenbrenner's idea that development and learning "is a function of the number of supportive links existing between the educational environment and other environments (such as home and family)" (p. 209) was incorporated into the notion of "resilience" by Twum-Antwi et al. [22] to design family-school interaction programs. Thus, the RSEC would be explicitly located within the mesosystem (school, home, community), although it extends beyond family–school relationships. This concept also considers other community agents to be relevant, such as community entities and institutions in which students participate and with which the school comes into contact (i.e., boys and girls clubs, local music schools, cultural and sports entities, etc.).

Several mechanisms have been proposed to operationalize the CSER, among which we wish to highlight three: "(a) orientation, social support and personalization; (b) identification and use of existing knowledge, resources and strengths; (c) collective action and culture of participation" [20] (p. 3). Although the authors of this formulation are interested in identifying these resilient mechanisms within geographical locations (communities, municipalities and even regions), we focus here on the educational practices of schools. These latter practices predate the pandemic and have been maintained, reactivated, or modified during the crisis, thus acting as resilience mechanisms [23].

The orientation, social support and personalization mechanisms refer on the one hand to practices related to the accompaniment and care of families (ample evidence exists regarding the fact that collaboration between families and schools is a key factor of academic achievement [24,25]). On the other hand, this factor would also include the creation of personalized learning trajectories based on the recognition that we learn within a multiplicity of spaces, contexts, situations and moments throughout a life cycle [26,27]. Both sets of practices can be especially relevant when dealing with attention to diversity and in environments of social exclusion, where there is a broad spectrum of individual needs.

The practice of identifying and using existing knowledge, resources, and strengths rests on the consideration that all families have vital experiences, accumulating and building knowledge, skills, resources, and wisdom. Potentially resilient practices linked to this factor would thus be those based on the identification of families' knowledge funds and students' identity funds as key elements to redesign academic activities in connection with students' living environments [28,29].

Finally, in terms of collective action and participatory culture, we can identify a wide range of practices that turn education into a complex activity that incorporates multiple educational agents in schools. Such practices include, among others, intergenerational learning processes—such as the University of California's Fifth Dimension model created to foster the development of vulnerable communities in their environment in collaboration with their students [30,31], and that has been implemented in other latitudes [32]; the diversification of the forms of relationship with families and their inclusion in school dynamics [33]; learning communities [34]; the participation of university students in service-learning programs [35]; and networking with other community entities [36,37].

All these practices represent different kinds of educational innovation that had been highlighted in the pre-pandemic literature as successful actions. Numerous practices have been implemented in schools in vulnerable environments and many are the result of collaborations with various community actors. If we understand the pandemic as an unexpected "experiment of nature" [31], we can identify which factors may be contributing to resilience in these schools. The disruption caused by this crisis and the adaptive response of educational institutions can thus help us to understand the development of practices that have demonstrated, over time, their ability to adapt to sudden and unexpected changes. We thus encounter, as described by Bronfenbrenner, "sudden changes in the ecology of development resulting from large-scale events" [21]. The objective of this work was twofold: first, to identify which ongoing educational practices performed by the vulnerable schools under study successfully adapted to COVID-19 changes; and second, how they can foster school interventions and transformations to achieve inclusion, equity, equality and sustainability.

## 2. Materials and Methods

The study we present here is part of a larger research project whose general objective is to identify the processes of educational change in vulnerable schools, analyzing their function as resilience mechanisms during the crisis caused by the COVID-19 pandemic, as well as their contribution to inclusion and educational equity.

Specifically, in this article, we present a qualitative case study analysis of two specific schools with socially vulnerable students in which continuous practices of educational innovation have been identified. Thus, the question we are trying to answer in this study is as follows: how have these innovative strategies been affected by the pandemic and, at the same time, to what extend have they served as effective tools to face the crisis and recover from its effects?

### 2.1. Context

We chose 2 schools from Seville and Barcelona for this case study because they had treasured a history of documented educational change processes for at least 3 years before the pandemic. Both schools are centers aimed at the educational inclusion of children belonging to Roma communities with high rates of failure and early school dropout.

The Seville school is a public infant and elementary school (children aged 3–12) where 90% of the children are of Roma ethnicity. This school faced high rates of classroom violence, absenteeism and school failure in the past; thus, the school decided to introduce changes in its organization. In 2006, with the agreement of teachers, principals and families, the school became part of the Community of Learners program [34]. A school that is part of the Community of Learners program implements a Freirean-based educational methodology, and the school then becomes a resource for the community that is open to all and used by all. The entire community participates in it—neighbors, parents, teachers, volunteers, university students and students of the center—and everyone learns together, participates in the classroom, has a voice and can make decisions. In these schools, the key to the educational process is to learn based on using egalitarian and horizontal dialogue [37,38]. This educational model allowed the incorporation of values and practices of the Roma community into the daily life of the school, all of them arising from the voices and actions of the children and their families. The teachers and other staff are in close and warm contact with all children and their parents, and they are in a relationship that is far removed from what is customary in traditional education. All this has made the school a very important resource for children, families and, in general, the entire community.

The Barcelona school is an infant, elementary and secondary school (3 to 16 years old); half of its student body is made up of the Roma community and the other half comprise immigrants from Africa, Asia and Latin America, and there is an absence of the native white population of the neighborhood. The situation of segregation together with the high rates of early dropouts from secondary schools in the neighboring high schools forced numerous changes in the school. It was transformed into a primary and secondary school (3 to 16 years old), organized into a community of practice and included in a network of neighborhood organizations. It also began a close collaboration with the university to develop projects such as the Fifth Dimension [32], the Funds of Knowledge [38], the Identity Funds [39] and the Community Funds of Knowledge and Identity [29]. Five years later, this transformation allowed the return of the population that previously avoided the center and substantially improved the permanence of the student body until the end of compulsory education at 16 years of age—all within a center with a great diversity of educational agents in addition to teachers.

### 2.2. Participants and Materials

In this study, semi-structured interviews were conducted with the principals of the two schools studied. Both were women in their fifties and had held office in their schools for more than five years.

The interviews and categories of analysis were elaborated using the tools designed for the broader research project these interviews were part of and based on a total sample of 29 schools. The interview script focused on information relating to the challenges as well as the resources employed at four different historical moments: pre-pandemic, lockdown, post-lockdown, and at the time of the interview. In the pre-pandemic period, we mainly asked about the school's characteristics and its general organization, to broadly understand the defining characteristics of these schools with vulnerable populations. Subsequently, we inquired about the difficulties they encountered and the strategies used to solve them, paying special attention to possible pre-pandemic actions that would have provided some crucial support to face the more challenging periods. That is, we sought to identify which pre-lockdown educational strategies could have served as resilient elements during the pandemic period and after.

### 2.3. Procedure and Analysis

The data collection and subsequent analysis was common to the entire matrix project.

Researchers who knew the schools conducted the interviews with the two principals to collect the data. The interviews lasted from one and a half to two and a half hours, and they were recorded with the consent of the interviewees.

We opted for a qualitative content analysis of a deductive–inductive nature [40,41] to produce a systematic description of the interview data. The interviews were transcribed using Amberscript 2.0 software and were then reviewed by the interviewer to ensure the transcription was correct. Next, each interviewer prepared a report highlighting the most relevant aspects of the interview. They obtained a series of emerging themes that gave rise to a set of topics for analysis, which, in turn, were organized according to their timing as described above: pre-pandemic, lockdown, post-lockdown and at the time of the interview.

The full interview transcripts were analyzed using the Atlas.ti 22 program. For coding and subsequent analysis, all interviews were divided into units with their own meaning, which usually coincided with the turns of speech. They were coded in different cycles until properties emerged making it possible to establish categories. The procedure was as follows: two researchers, one of whom conducted the interview, coded each interview independently, classifying the data according to the research objectives. These categories were redefined in the different coding cycles. The category system (Table 1) was tested by coding 4 interviews, each conducted by two independent researchers. The categories that emerged from each pair were compared until an agreement was reached on the definition of each category. Once any inconsistencies were adjusted, a final category system was generated and contrasted by re-coding and confirming that each of the four pairs of coders reached above an agreement rate of above 90% for each interview. Finally, two researchers coded each interview in the overall study. The results of two of those interviews are presented below. They correspond, as noted above, to schools that had implemented outstanding innovative practices.

**Table 1.** Category system.

| Moments | Categories |
|---|---|
| Before COVID-19, (Beginning 2019–2020 academic year) | School characteristics and inclusion strategies |
| | Pedagogical innovation |
| | Family–school connections |
| | Teacher's situation and conditions |

**Table 1.** *Cont.*

| Moments | Categories |
| --- | --- |
| During lockdown (March–June 2020) | Teaching organization |
| | ICT employment by teachers and pupils |
| | School tasks and family organization |
| | Care and help for students |
| | Communication channels with families |
| | Care and help provided to families |
| | Coordination with other services (Social Services, NGOs . . . ) |
| Post-Lockdown and now (academic year 2020–2021) | Incorporation of new methods and contents in teaching |
| | Restructuring school organization |
| | Curriculum adaptations and motivating strategies for inclusion |
| | Care and help for students |
| | Psychosocial problems in families: care and help provided |
| | New needs found in communities |
| Interview moment (academic year 2021–2022) | Consequences that COVID-19 has left, how they continue to affect and how they are faced |
| | Medium- and long-term consequences in student's learning processes |
| | Pending issues for inclusive education. Issues to improve |

## 3. Results

The following are the main results obtained after an analysis of the interviews with school principals of the Barcelona school (BS) and Seville school (SS) (names of schools are omitted). These results were grouped into four different moments (before COVID-19, confinement, post-confinement and the moment of the interview (21–22)), following category system shown in Table 1.

### 3.1. Before COVID-19: School Organization, Characteristics and Inclusion/Innovation

Regarding the educational practices oriented toward inclusion and innovation that were already in place in the schools, there were several elements that both schools pointed out as crucial in their day-to-day life prior to the pandemic, and which they also pointed out as basic elements for dealing with the difficulties they encountered during the pandemic period.

In the first place, in both schools, there were mediating figures, such as social educators, who established a direct connection between the school and other social institutions. Secondly, in both cases, economic aid was available, and this included allowing students to eat at school free of charge. Thirdly, the teaching staff of both schools is stable. These schools, due to their special characteristics of having a vulnerable population, have a teaching staff that does not change constantly since they request specific assignments to these schools. As the fourth point, both schools are categorized as communities of practice (Community of learning in Seville and various communities that provide group courses in Barcelona), and this has two clear advantages: On the one hand, there is flexibility when teaching, thanks to the interactive and collaborative groups that allow advancements toward new teaching structures with diverse roles. On the other hand, the second advantage is that these schools exhibit a very consolidated family–school relationship, and families understand the school as a resource and support structure that is not only an academic entity but also provides social and even economic resources. As a fifth point, these schools are immersed in ICT promotion programs in both cases, especially in Barcelona, because that school was a pilot center for implementing digital educational strategies with a re-

cently created app intended for fostering interactions with families. Finally, in both centers, intense collaboration with other entities within their environments occurs (public library, civic center and music school in Barcelona and student residence in Seville—both have a university community via Service Learning programs). The principal of the Barcelona school refers to how these collaborations and previous innovations placed the school in an advantaged position to deal with the challenges required by COVID-19:

> *"This means that these ways of educational transformation that you have mentioned, all of them have been part of our zero moment at the beginning of the project, in such a way that we lost our fear, we lost the fear of being a complex center and we lost the fear of innovating in complex environments".* (interview BS)

*3.2. During Lockdown: Difficulties and Strategies*

During the lockdown period (March to June 2020), the schools encountered numerous difficulties that they had to face. The main ones are as follows.

In order to organize teaching, the schools needed their students to have basic school materials at home, which was not previously the case, and all academic work fell on the students, with little support from a large part of the families. School materials for community use were housed in schools. In the case of the Seville school, the students do not work with books, which meant that they could not take home support materials. In the case of Barcelona, thanks to the coordination with the "Pla de Barris" of the Barcelona City Council and the funds from different institutions in the case of Seville, families were provided with basic school supplies: notebooks, pencils, paper, etc.

The fundamental challenge was to establish and maintain communication with students using digital media when many families lacked them and did not own devices beyond a cell phone, which in many cases was reduced to one cell phone per family. Thus, both schools addressed these shortcomings by providing families with available computers and tablets, in addition to negotiating with the corresponding educational administrations for obtaining more devices. This was key to maintaining contact with families, with different results in this case for the two schools. Thus, while many families in Seville did not know how to handle the devices or even encountered problems during video conferences, the Barcelona school already had an app for communications that the school sent to the families before the confinement, so it only had to be adapted to bidirectional applications.

In any case, as long as the main objective was that no family was left isolated, the intense use of phone calls in order to reach families was necessary in both schools. The teachers provided their personal telephone numbers and 24 h availability for the families in most cases. In addition, the calls were sometimes taken advantage of by the siblings of the instructed children, and they sometimes turned into inter-level sessions. The following statements illustrate this situation.

> *"It didn't connect every time you called it, they didn't answer, and you had to insist again".* (interview SS)

> *"The school class was given through WhatsApp video call ( . . . ) and we knew about some families because some children told you: Yes, teacher, my cousin . . . ".* (interview SS)

Videoconferences, in addition to being tools for connecting with families, served as a support for school activities. In fact, this was the only way to work with students in early childhood education (3 to 6 years old). In addition, in the Barcelona school, the use of the Google Classroom platform in primary levels (6 to 12 years) and especially in secondary levels (12 to 16 years) served as a method for structuring activities. The principal of the Barcelona school said the following.

> *"We had 15 tablets, 80 computers. We made them available to the families, obviously in order of student, from oldest to youngest, so that they could progress . . . . We had started to work in the classrooms with the Classroom app, but we had just started, and suddenly that was already our working platform. And at a younger level, through this app we set weekly tasks that families could send us, even with a photo".* (interview BS)

As this period progressed, new problems were detected, such as the lack of continuity on the part of the students with respect to the completion of homework. This complicated the follow-up study; as a result, academic performance worsened significantly.

In addition, from the beginning, many families were suffering from severe isolation that not only introduced physical but also social effects. This situation generated emotional problems that, in some cases, translated into an increase in violence within homes. These families, in trying to survive, found themselves unable to help their children with academic tasks and, in many cases, turned to the school for help. All of this led both schools to an adaptation of curriculum objectives, prioritizing essential course content along with the incorporation of objectives related to social connectedness and emotional care for these children and their families.

During this period, schools responded and functioned as a community resource that was not only academic in nature and provided organized and centralized basic needs such as food donations in the neighborhood, but the schools also served as emotional support structures. We illustrate this with the words of the principal of the Seville school.

> *"Yes, it was a shock for many families. In general, it was very hard because many of our families make a living every day. Then they were looking for a living by parking cars, looking for recycling items in garbage, and their economic means disappeared. It is true that education is very important, but education for our families went to level number . . . I don't know which . . . , because they say "and now how do I eat: there is no school canteen, I can't go out to make a living, I don't have money . . . " Therefore, we became resources and support to Social Services, and we were included in an emergency commission that was formed in the neighborhood. We provided data on families that we knew were in a situation . . . that is to say, hey, this family should be given some pure and simple food aid, because they have to eat, they are not going to go shopping".* (interview SS)

*3.3. Post-Lockdown: Difficulties and Strategies*

During the post-lockdown period, essentially the 2020–21 academic year when the schools reopened, the situation was particularly delicate in these vulnerable social environments, forcing new re-adaptations and changes.

In the return to face-to-face attendance, the fear of infection by families and absenteeism was the biggest challenge these schools had to face. To avoid infection, numerous measures were taken, such as adapting the different regulations issued weekly by the administration to the situation in the community. For example, guidelines related to the selective confinement of infected persons and close contacts—from an assumption that all families were nuclear families—were sometimes adapted to the conditions and practices of Roma families, such as those living in extended family groups. The following statement accounts for these adaptations.

> *"Look, no, with most of them we did not do it (account for the absenteeism of some children), because afterwards it is true that there was a lot of fear to bring and share. We saw that fear, but you saw the child in the street. Yes, we understand what this fear comes from, of course, that "my family protects him; the child belongs to the family and is protected from a virus". And at other times, we did it with some boys and girls (leaving them in isolation and implementing tele-training even though they were not in close contact, following the nuclear family rule). We did it although they did not have all the papers that we were asked for so that they would be exempt from coming to school in a justified way"* (acknowledging that the extended family was close contact for these children). (interview SS)

Likewise, new health and safety rules were established, and both schools made an effort to make it clear that they were being followed. Personal communication with families was maintained in order to build trust and retain collaboration with respect to the children's development. In this line, face-to-face collaboration with universities was canceled to reduce the number of staff in the center in order to maintain the level of trust

held by the parents that their children would be less likely to become infected by their teachers compared to the risk of infection that they associated with the university students (based on the families' perception of the danger that contact with young university students entailed). We illustrate this in the following statement.

> *"The same sense of security that led them to feel no danger in being with the family in the small square (shared community space in the streets), but they were afraid to take the children to school. As they felt the teachers were close to them, they did not feel danger. However, they did not consider including the young university students, because they were perceived as a source of danger".* (interview SS)

Another measure taken after returning from confinement was to avoid being inside the school for long periods, and this required extending activities into open-air spaces. In this sense, external classrooms were created, and subjects related to environmental education were introduced. All this contributed to these schools being understood as safe spaces where families could leave their children in complete confidence. Therefore, during the period of confinement, there was a clear need to maintain mutual trust with families by using forms of interaction in which affective aspects played a relevant role. The principal of the school in Barcelona told us the following.

> *"Yes, but I have to say that it was once again to appeal to trust and to appeal to the way we work so that they understand that the school is a safe place. Why? Because we, regarding the COVID could not assure that the school was a 100% safe place, which was the fear they had, but we did show them that, as we were very intransigent, with positive look, regarding the performances that were done. Everybody had open windows, everybody put on gel, washed their hands, had their masks on at all times, groups that did not mix. We were very careful, more than careful with sanitary measures. We reached a very low level of contagion from the beginning, so that helped to generate trust. And since there was also an inner desire to return to normality, the way we acted so well made the families give us a vote of confidence and the children quickly returned to school".* (interview BS)

The return to school showed that students presented considerable delays in understanding the curriculum, exhibiting lower academic performance than before the pandemic. This, moreover, was accompanied by an increase in mental-health-related problems (especially among secondary-level students). Thus, differentiated needs emerged in each age group, which required individualized curriculum adaptations. In the Barcelona school, in order to favor physical distancing within classrooms, students were distributed in more classrooms: every two classes occurred in three classrooms. Thus, for example, fifth and sixth grade students (aged 10–11) were grouped and distributed in three new mixed groups so that there were fewer students in each group; however, there were greater differences with respect to learning levels within each class. The previous organization of classes as communities of practice allowed the development of tools that now proved useful for this new multilevel organization. The organizers said the following.

> *"Regarding the forms of cooperative work, we, with a pedagogical perspective . . . , we set ourselves from the first moment to generate different spaces. We began to work as communities. At the beginning, they were organizational communities, but they ended up being learning communities. Moreover, after the pandemic, we started working as multilevel communities, not without hesitation, but we started working as multilevel communities. This means that we have children of different ages in the classrooms".* (interview BS)

In contrast to the difficulties in other areas, after the improvised use of new digital tools during the lockdown, the return to classrooms occurred with improved competencies with respect to the students and teachers. This made it possible to increase activities that use these tools and also allowed the improved identification of the training needs of teachers in this regard.

A clear setback was observed with respect to the participation of students' relatives and people from collaborating entities in the daily activities of students (such as the university, always with the aim of offering security to the families). The same occurred for the students' activities outside the school. Even so, the relationship with all these entities was maintained in order to facilitate their recovery as soon as pandemic measures permitted it.

*3.4. Interview Moment: Difficulties and Strategies*

The last period studied corresponds to the "normalization" period during the middle of the 2021–2022 school year. Special pandemic measures—such as there being separate groups, the use of masks or the restricted access of people other than students and teachers—were progressively deactivated.

However, this "normalization" is relative, since some of the after-effects that were already visible when students returned to the classroom during the previous year are still present (and even intensifying). In general, a problem that continues to be detected in both schools is a regression of social skills, which in turn generates conflicts and violence, especially among older students. After-effects have also been detected by age groups. In the case of infant education, problems were related to the area of language, such as delays in its acquisition in 3-year-old children or situations of selective mutism. Delays in motor skills and emotional problems have also been observed in these children, and these observations include difficulties in establishing bonds with teachers or the fear of physical proximity. In primary school students, problems have been observed in the acquisition of reading and writing skills, as well as in study habits. However, it is in secondary schools where more alarm has been generated, as mental health problems have intensified. Thus, increases in self-harm, anxiety crises, depression, isolation, conflict and harassment via social networks have been reported. It has even been found that at the time of the removal of the obligation to wear a mask, several girls were reluctant to show their faces. Although adolescents are the group in which these consequences have been most evident, an increase in mental health problems has occurred in students, families and teachers as a whole. In the following statement, we can observe how the principal of the Barcelona center outlines the problem for the case of infants and primary education.

> *"Well, we have observed different types of affectation at the mental health level, because, on the one hand, we could see in young children a lack of stimulation from home, that is, everything that from school is worked on. I am not talking about knowledge, I am talking about experiences, habits, limits, routines, and, in some way, when they (the children) are incorporated at school, and these gaps are detected. Moreover, referring more to knowledge and experiences, which would be more primary, well, they have originated these mental health issues, of crises more at a behavioral level, of contained rage, of little empathy in the relationship with others".* (interview BS)

This return to "normality" meant the withdrawal of additional human resources provided by governmental institutions during the pandemic, with the result that some of the measures adopted can no longer be implemented, and this increased the workload for the teaching staff.

> *"That is, the office work has tripled by 100, 300, 1000 . . . I, for example, today I have not opened the mail and I do not feel good. I am not ashamed to say it because you have already seen me, you saw me coming from a meeting, but here I have been in two classrooms, I have solved a couple of situations, I have had two meetings".* (interview BS)

Despite this, schools continued to implement strategies to alleviate these situations. In response to the social and emotional difficulties detected in the students upon their return to the classroom, tutorials had been organized for individualized support, and emphasis has been placed on cooperative work for group cohesion. The following observation is provided.

*"This year we participated in a mental health pilot project given our complexity. Obviously, this mental health pilot project involves all the services from health and education. And we meet once every 15 days. And for us that gives us stability, security, and helps us plan how we want to work with these children".* (interview BS)

Teamwork and digital tools remained in school planning and organization, with teachers meeting and sharing documents online, which allowed for improved coordination. Likewise, some forms of group and space organization that proved to be practical were maintained: grouping by age groups during playtime, which was mandatory during the year after the pandemic, was maintained afterwards, as it was found to reduce the chances of escalating violence.

Something that remained with the end of the pandemic crisis was the strengthening of ties with the community. Regarding families, the "new relationship" based on collaboration and trust promoted by the mediating role assumed by the school during the lockdown was recovered in the following periods. This contributed to school life, and teachers continued to be perceived as part of the community. In the following examples, we show this process.

*"As resilience, I would like to think that somehow the relationship we have had with families has been . . . , we have reinvented relationships with families, and that can help us. We have improved the relationship with families and for me that is very important and we must make an effort to maintain the relationship between students, between teachers and students and teachers".* (interview BS)

*"Sure, just as we gave our phone, many teachers gave their phone in the lock-down to parents. Moreover, it is that parents call them to tell them and communicate anything, at any time and at any moment. Therefore, this fluid communication channel is already in place. I think that parents already have any problem and they communicate with us".* (interview SS)

Finally, previous programs that involved the participation of families were recovered, such as the Learning Community program in Seville, the Funds of Knowledge program in Barcelona and Service Learning with universities in both schools; in addition, the relationships with surrounding entities were recuperated, and were strengthened as the school emerged as an important community resource.

In general terms, these schools found that the pedagogical innovations undertaken before the pandemic placed them in a scenario that allowed them to generate resilient strategies for facing difficulties that emerged from the pandemic, reinforcing their pedagogical and community values and practices. This was especially salient in relation to the care and promotion of the integral development of students beyond academics, with exceptional emphasis on affective and social aspects as well as the conviction that only within cohesive and participatory communities can this growth take place. This is how our interviewees expressed it. We want to end this section using the voicing of their words.

*"So I believe that the importance of human value has been highlighted. If it has ever been thought that robots are going to replace teachers, I believe that the pandemic has shown that a robot does not replace the teacher because of the human part . . . maybe in 80 years, but I don't know".* (interview SS)

## 4. Discussion

The present study focused on two schools that serve socially vulnerable populations and that have continuously implemented innovative educational practices. The interviews shed light on educational system needs that surfaced during the pandemic and which have been widely established by other authors over the past few years, e.g., [5–12]. Nevertheless, the present study delved further: we found that despite a high degree of improvisation, the strategies implemented by these schools to overcome the crisis were largely based on educational change processes that had begun before the crisis. These previous inclusion strategies lay at the heart of the schools' resilience and allowed implementing measures

that we can group into five domains, some of them closely related to "Community Socio-Educational Resilience" (CSER) mechanisms [20].

The first domain is related to the organization of communities of practice in schools. With models such as the Fifth Dimension [30] or Learning Communities [34,42], schools transform traditional practice and perform a paradigm shift. They move from a transmission-centered model onto a collaboration-based model [43] and eliminate the teacher figure of the sole transmitting agent before a group of receiver-students, thus legitimizing new educational figures. Such models therefore turn education into a complex activity with multiple educational agents (families, volunteers, social educators, specialists, leisure time monitors, university students in service learning programs, etc.). The interviews revealed the relevance of organizing "communities" in schools that facilitate, among others, the co-ordination of various actors, teachers, and community members when developing learning activities. In the interviews, this school organization shed light on the importance of having a stable teaching staff. Scarce teacher turnover was observed. Indeed, these positions were specific destinations chosen by the teachers themselves who wished to be in these schools owing to their special attention to vulnerable populations. Finally, we must emphasize that this way of understanding education and of configuring schools emerged in the interviews as a key factor that facilitated the implementation of other strategies, detailed below.

One of these strategies corresponds to the second notable domain: relations with families and new forms of family participation. A considerable body of research shows that collaboration between families and schools is a key factor in academic achievement [24,25]. It has also been emphasized by Twum-Antwi et al. [22] as one of the key resilience factors when a crisis occurs. During the lockdown, family members had to be the educational agents for children who were immobilized at home, so their involvement was more relevant than during normal periods. Thus, relations with families played a leading role during all phases of the pandemic, strengthening internal communication and creating new channels of communication and coordination. As the results showed, this was possible, on the one hand, because both schools had a well-established family–school relationship and, on the other, owing to the existence of mediating figures, such as social educators, who had already established a direct connection between the school and other social institutions. During the confinement period, both schools had to resort to telephone calls to communicate directly with the families, and in both cases, teachers provided their personal telephone numbers and offered 24 h availability. Another facilitating aspect was that the schools' status was acknowledged by families from the very start: they understood that the schools were not only an educational resource for their children, but also a community resource. This image was consolidated during the pandemic, especially in the case of highly vulnerable families and at risk of extreme poverty. The crisis required that schools act even more as a social service provider, becoming true community agents and regarding the well-being of families as a priority. An illustration is the fact that they provided catering services to a large part of the schools' students and their families. Not only was material aid provided, however: another major problem identified after the pandemic was the severe isolation suffered by families. This situation generated serious emotional problems, which, in some cases, resulted in an increase in violence within the home. The generated trust in the school led families to regard them as the most accessible resource to turn to for help and support.

The return to normality was also a challenge for schools, as families were highly reluctant to come back and feared contagion. This resulted in an increase in absenteeism, which became the schools' biggest challenge. Some of the adopted measures included communicating with families to regain their trust, suspending any access to personnel from outside the school, or adapting the rules dictated weekly by the administration.

The third notable resilient strategy domain was the personalization of teaching and the use of existing knowledge, resources and strengths. This new understanding of learning recognizes the multiplicity of spheres, contexts, situations, and moments throughout a life cycle in which we learn [26,27]. It also considers how all families, in all circumstances, have life experiences, knowledge, skills, abilities, resources and wisdom, as shown by

the Funds of Knowledge educational project, which incorporates skills and knowledge drawn from family practices while strengthening the relationship between school and families [28,29]. In short, the issue is to move away from a uniform conception of education, a requirement that is particularly visible when dealing with attention to diversity and social exclusion environments that encompass a very broad spectrum of individual needs. The lockdown introduced this precise, exceptional situation, where not all students were attending, thus highlighting the diversity of their individual experiences. The interviews conducted showed us the schools' difficulties at implementing home-based education, precisely because of this variability, and because of the basic deficiencies they suffered—students did not have the necessary school materials or digital media to follow the lessons. In addition, a lack of work habits at home and an absence of continuity in homework completion generated a decline in academic performance during the lockdown. This finding supports the conclusions of different studies regarding the highly negative effects of the pandemic on both students' academic results and mental health [44], triggering inequalities within the classroom. Moreover, the latter findings especially apply to vulnerable social environments [45,46]. Such difficulties put the need for personalized student attention back on the table: students manifested very different needs within each class, as well varying levels of progression during the lockdown, and distinct experiences. For this reason, it is necessary to understand students' trajectories, beginning with their family environment and significant models. The schools analyzed showed personalization precedents through knowledge of family dynamics, such as that promoted by the Funds of Knowledge and Identity Funds at the Barcelona school or the Learning Communities in Seville. Thus, differentiated needs arose that often required individualized curricular adaptations, though more generalized adaptations were also introduced for the whole school. The return to the classroom meant implementing flexible mechanisms that would allow the recovering of previous academic levels, as well as normalizing the teaching processes, sometimes prioritizing contents related to emotional attention, which had been so depleted during the confinement period. Both schools' previous experiences provided considerable advantages for the recovery of normality. This was observed in the case of the Seville school, which benefitted from its interactive and collaborative group structures, gradually returning to the collaborations with volunteers and university students that lay at its heart. In the case of Barcelona, in order to favor social distancing within classrooms, the courses were merged to create three, smaller multilevel groups. This new organization was facilitated by the existence of a previous structure that had involved coordination and joint planning. The positive outcomes achieved have led to maintaining this organization to this day, giving rise to a process of specialized teacher training in multilevel teaching practices. Finally, the schools' pre-existing flexible structures also facilitated the expansion of outdoor activities, creating external classrooms and introducing new subjects, among which some related to environmental education.

The fourth area of emphasis focuses on the continuity of school practice via digital media. Since the 1990s, the introduction of new technologies into schools has been accompanied by a digital divide that has added to other social inequalities [47]. The lockdown required different forms of distance education, accentuating the pre-existing digital divide even further [48]. This fact was highlighted in the schools analyzed, as it was reported that families had to be provided with devices to keep them connected. We commented earlier on the key role of these devices to maintain communication with families, but beyond this, they also served to maintain the continuity of school activities. Regarding this point, we detected a notable difference between the Seville and Barcelona schools. The latter school had previously led active programs both to advance the use of digital tools within the school and to communicate with families, providing them with a head start. During this period, it became clear that overcoming the digital divide required the inclusion of families and the provision of material resources and skills that could be integrated into meaningful practices. In addition, online work meant progress in teaching organization and planning, which was subsequently maintained. In any case, the return to the classroom

was accompanied by an awareness of the need to maintain the ICT achievements during this period, highlighting the need to expand ICT teacher training, as well as the maintaining of online activities and digital communication between the different school community agents [16,17].

Finally, a fifth domain deemed fundamental for the advancement of schools was the offshoring of school practices. The latter refers to the importance of creating educational networks based on alliances between schools and other agents such as libraries, museums, cultural entities, music schools, artistic-creation centers, leisure centers, commercial entities, etc. This transforms the school from a hermetic entity disconnected from its environment into an educational network node, expanding the time, space and diversity of educational processes and recognizing the multiplicity of learning situations [49]. The results showed us how the openness and connection that both schools already enjoyed with their community environment were crucial to face the pandemic challenges. As mentioned above, during the lockdown, these schools functioned as community centers to distribute resources such as food and digital devices or to connect with social services—and this was thanks to the pre-existing networks that allowed the coordination and rapid flow of information and resources. The return to normality meant recovering programs and networks that had been put on hold, but which were the basis for a rapid recovery of what had been lost. Thus, the Barcelona school referred to the importance of its relations and activities with the public library, the civic center or the music school, and the Seville school its relations with the university or the student residence—from which they recovered the Service Learning programs [50] that allowed them to implement a part of their educational strategies.

To summarize, we found that these five domains played a fundamental role in educational resilience. Especially notable was the relevance of the pre-existence in these schools of educational models that enabled the implementation of many of the actions described above. The networks established by these schools with the community, the relationships with families, the incorporation of external agents in the educational process, personalized attention, social support, all support the concept of "Community Socio-Educational Resilience" (CSER) [20]. To the above we must add the importance of introducing digital tools as fundamental resources to maintain relational and educational processes.

## 5. Conclusions

The study presented here illustrated the countless challenges and adaptations that professionals and families linked to vulnerable schools had to face during the COVID-19 crisis. We identified resilience mechanisms in the practices of these schools that helped educational systems to address the impact of the pandemic. The main mechanism consisted in the types of relationships established by the schools with families as well as with the community. These strategies were facilitated by each school's pre-existing structures and forms of organization.

The main challenge faced by schools that serve a mostly minority-group population is the discontinuities between the school and family practices and values. Hence, programs that aim to create spheres of continuity play a key role, as observed in the Learning Communities or Funds of Knowledge programs. The results of the study suggest that the prolonged existence of these programs provided school communities with resources that allowed them to face the crisis and to recover after it ended. The main challenges of vulnerable schools are the families' needs for social and emotional accompaniment, which involves strengthening their community dimension. Families need to perceive schools as safe areas in which collaboration and flexible relationships of mutual trust are woven.

In addition, the study brought to light other elements that also contribute to educational resilience: the personalization of teaching, the meaningful use of digital tools, and the notable role of community networking as fundamental strategies to move forward after the crisis. We also found that digital mediating tools played a decisive role in periods of crisis, such as that of the pandemic.

Regarding the theoretical implications of our study, the inclusion of these elements can contribute to completing the concept "Community Socio-Educational Resilience" [20]. The school plays a fundamental role in vulnerable social environments: schools provide affective security as they care for the built relationships; they legitimize cultural differences through practices that link the community world to the school world; and they include students and their families within their institutional framework and connect them to a social network. Crisis situations such as the pandemic have helped us to identify resilient practices in an environment in which adversity often presents itself in the form of successive and varied crises.

## 6. Limitations

To finish, we wish to point out that the main limitation of this case study was the single category of actors represented in the sample: school principals. The matrix research project underlying the present work is designed to complete this initial evidence via a more complex study that includes participant observations and interviews with teachers and families, as well as discussion groups with students from different primary and secondary schools. We thus hope to consolidate these results. Nevertheless, we should also point out that since this was a qualitative study based on a small sample, it would be inappropriate to generalize the results. What we present here is rather a starting point for future studies to analyze resilient processes and the changes that the latter have brought about in schools with vulnerable populations in their daily work of attending to diversity and inequality. This subject should be pursued in future studies. In this line, we have extended this interview model to another 27 highly complex schools in six Spanish cities, and we have explored the subject more in depth in 10 of them through ethnographic work. We hope to present the results of this latter research shortly.

**Author Contributions:** Conceptualization, V.M.-L., B.M.-G.-E. and J.L.L.; Methodology, V.M.-L.; Investigation, V.M.-L., B.M.-G.-E. and J.L.L.; Data curation, B.M.-G.-E.; Writing—original draft, V.M.-L. and J.L.L.; Writing—review & editing, B.M.-G.-E.; Project administration, V.M.-L.; Funding acquisition, J.L.L. All the authors have contributed their work to all sections of this paper, from research to the writing of each of its sections. All authors have read and agreed to the published version of the manuscript.

**Funding:** This research was funded by the MINISTERIO DE CIENCIA E INNOVACIÓN, SPAIN, 2020. Grants corresponding to the 2020 call for "I+D+I" projects within the framework of the State programs for knowledge generation and scientific and technological strengthening of I+D+I system; and I+D+i oriented to the challenges of society. Reference: PID2020-118198RB-I00.

**Institutional Review Board Statement:** The study was conducted in accordance with the Declaration of Helsinki, and approved by the ETHICS COMMITTEE ON ANIMAL AND HUMAN EXPERI-MENTATION (CEEAH) OF THE UNIVERSITAT AUTÒNOMA DE BARCELONA (reference number CEEAH6272, date of approval 2/12/2022).

**Informed Consent Statement:** Informed consent was obtained from all subjects involved in the study.

**Data Availability Statement:** Not applicable.

**Conflicts of Interest:** The authors declare no conflict of interest.

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
