# Peer review of "Community Resilience Processes in Schools with Roma Students during COVID-19: Two Case Studies in Spain"

_sustainability, doi:10.3390/su151310502_

Round 1

Reviewer 1 Report (New Reviewer)

The authors of the manuscript entitled “Community resilience processes in schools with Roma students during COVID-19. Two case studies in Spain” have revised the content of their work. In my modest opinion, revisions have addressed most of the concerns that reviewers expressed during the first round of reviews. Substantial changes have been made to the introduction to elucidate the theoretical framework of the study, to the method section to clarify the procedure used by the researchers, and to the discussion. The latter now adequately contextualize the study. Besides a modest level of proofreading that may enhance the structure of the authors’ writing and readers’ understanding of the study, all contents point to a successful revision of the original manuscript.  

See text above.

Author Response

Attached cover letter,

Bests

Reviewer 2 Report (New Reviewer)

Thorough processing of the impact of covid on education is very important for future similar situations. The topic of the article is beneficial and can complement the mosaic of already existing studies of this focus.

The characteristics of both schools are sufficiently detailed. This fact helps the reader better understand the context of the entire research.

The authors state that they present the results of the analysis of interviews with two school principals. However, in sections 2.2 and 2.3 they talk about 29 interviews. They mentioned above that this is part of a larger study, but since they don't report the results of the other interviews here, the mention of 29 interviews is confusing. The reader can expect an analysis of 29 interviews.

Duplicate coding increased the accuracy of data analysis.

The description of the entire process before, during and after covid is detailed. Authors should consider whether long quotations from interviews are effective.

Author Response

Attached cover letter,

Bests

Reviewer 3 Report (New Reviewer)

The paper studies an interesting topic and prospect for further is provided. The manustript is well-organized and structured.

-It's research purpose is clearly stated and appropriate method of study is applied.

- The thoritical contribution of the work is improved and relevand.

-The results have taken in to account the objectives of the research.

-The disussion is more efficient and responds to the results.

-There is more cohensive connection between the conclusions and the discussion.

Author Response

Thanks for your review,

Bests

Reviewer 4 Report (New Reviewer)

...

Author Response

Attached cover letter,

Bests

This manuscript is a resubmission of an earlier submission. The following is a list of the peer review reports and author responses from that submission.

Round 1

Reviewer 1 Report

The article describes the coping strategies to the difficulties generated by the Covid-19 pandemic in two Spanish schools with a high proportion of immigrant and Roma children. A case study provides interesting, yet quite predictable results from the viewpoints of the schools’ management teams. 

Introduction presents a theoretical framework of the study. Previous studies emphasize non-English literature which makes it hard for an English-speaking reviewer to evaluate the references. Also, self-citation of the third author was evident (9 references out of 49 equals every fifth reference!). The authors are encouraged to enwiden the theory section by adding new, international references especially in Personalization of teaching (p. 3, lines 137-149) and Relocation of school practice (p. 4, lines 164-176). Also, decrease of non-English literature as well as self-citations should be carefully considered.

Materials and Methods section provides an overview of the research data. More information is needed for the reader to evaluate the implementation of the research, e.g., how many interviews were implemented in this case study? Who where interviewed (i.e., who belonged to the school management team)? Also, according to an abstract, a manuscript describes a case study. This should be elaborated more in detail, e.g., why was a case study method employed in this study and how did it affect the research process? Further, it seems from the results section that group interviews were held. Please verify why and explain how they were implemented. Data analysis method needs also to be explained with references to research literature. What was the research question that was used as a basis for data analysis/classification?

One fundamental concern is related to research ethics and protecting the anonymity of the participants in the study. Providing the occupations and the schools names in the text let the reader to figure out the participants and, therefore, reveal them. In order to follow international research ethics, the authors need to remove all information that can be linked to the participants, especially schools’ names. Instead, the schools can be, for example, numbered and referred to their numbers in the text. Also, the authors should carefully consider what is the extra value of providing the city names. The city names could also be deleted and, instead, described only the necessary information related to them (e.g., the size, location and/or cultural heterogeneity of the cities) in the text. The authors should remember to target the text to the international readers.

Results section has a clear structure and provides descriptions of the authentic views of the participants. The numbering of the sub-titles in the results section needs to be corrected.

Discussion section returns back to the results. Discussion section could be stronger if the results are compared with previous research. Also, viewpoints and definitions of resilience and sustainability could be stronger since they are in focus according to the title of the manuscript and the Special Issue. Discussion could also describe limitations of the study.

Results and discussion do not support the conclusions drawn (p. 15, first paragraph). Instead, the study supports, e.g., the notions that a) relations with families as well as community are crucial during exceptional times and after them, and b) these can be supported by trust, personalization, flexibility and collaboration, including providing the families with digital tools and skills for their use. Conclusions should be closer linked with the main results of the study.

Finally, there are some minor comments to refine the text:

-          Keywords: ’culturally sensitive education’ and ‘community engaged research’ are not referred to in the text. These keywords should either be deleted or described in the text.

-          p. 1, line 10: Covid-19

-          p. 1, line 36: references 5, 6

-          p. 2, line 71: Covid-19

-          p. 2, lines 75-78: If I understand right, the family-school partnership is phrased twice in the list, first, as the family-school partnership and, second, as the diversification of the forms of relationship with families and their inclusion in the school dynamics

-          p. 5-6, Table 1: Beginning 2019-2020 academic year, 2020-2021, 2021-2022

-          p. 6, Table 1: TIC – explain the abbreviation

-          p. 6, Table 1, Post Lock-down: delete fullstops at the end of the Specific Interest Focus

-          p. 6, line 276: delete fullstop

-          p. 9, line 417: … rule). We…

-          p. 13, line 605: (a) In the … (add a space)

-          p. 15, line 719: In general terms, visibility…

Author Response

Dear reviewer, 

thanks for your careful review. The changes proposed clearly help improving our manuscript. Bellow I am enlisting the changes made to our manuscript, following your suggestions:

Abstract has been revised to ensure it includes purpose, method, key findings and implications.

The introduction has been reorganized into three sub-sections, with the purpose of clarifying the theoretical referents in three areas. The first two subsections have to do with the state of the art and the last one with theoretical development based on team practice. Accessible sources/references in English have been included, counterbalancing the excessive bibliography in Spanish. Self-references have also been reduced to those strictly necessary (when we refer to the background of this line of research).

The method has been revised to clarify the relationship of the study with its context, the analysis procedure, and the referents on which it is based.

In the results, the name of the schools has been anonymized, although the reference to the cities in which they are located is maintained, as it may be relevant for researchers for whom the differential local elements of the Spanish educational system are important, managed in a regional level.

The discussion has been expanded to connect more explicitly with the theoretical elements presented in the introduction, developing especially the issues related to community resilience.

The conclusions have been simplified with a new wording, adding more explicitly the limitations of the study.

Finally, some typographical errors have been corrected and an external English revision has been carried out.

Change control has been activated so that there is a complete record, except for the change of position of two paragraphs to prevent them from appearing as new contributions.

Bests

Reviewer 2 Report

I think that this paper has merit especially because of its high quality of structure and clarity. I really enjoyed reading this paper, which is based on a very interesting and significant theme. It is very well structured and has an appropriate methodology. It can be understood by a wide audience. I think that the paper could be published in its present form. It could also make a contribution to the dialogue regarding educational strategies for vulnerable schools in periods of crisis. It is also a paradigm of sensitive and engaged social and educational research.

Author Response

Dear reviewer, 

thanks for your careful review. Bellow I am enlisting the changes made to our manuscript, following all reviewer´s suggestions:

Abstract has been revised to ensure it includes purpose, method, key findings and implications.

The introduction has been reorganized into three sub-sections, with the purpose of clarifying the theoretical referents in three areas. The first two subsections have to do with the state of the art and the last one with theoretical development based on team practice. Accessible sources/references in English have been included, counterbalancing the excessive bibliography in Spanish. Self-references have also been reduced to those strictly necessary (when we refer to the background of this line of research).

The method has been revised to clarify the relationship of the study with its context, the analysis procedure, and the referents on which it is based.

In the results, the name of the schools has been anonymized, although the reference to the cities in which they are located is maintained, as it may be relevant for researchers for whom the differential local elements of the Spanish educational system are important, managed in a regional level.

The discussion has been expanded to connect more explicitly with the theoretical elements presented in the introduction, developing especially the issues related to community resilience.

The conclusions have been simplified with a new wording, adding more explicitly the limitations of the study.

Finally, some typographical errors have been corrected and an external English revision has been carried out.

Change control has been activated so that there is a complete record, except for the change of position of two paragraphs to prevent them from appearing as new contributions.

Bests

Reviewer 3 Report

The paper describes the coping strategies to the challenges generated by the COVID-19 pandemic in two schools which serve children mainly from immigrant and Roma families in Spain. The topic of the paper is extremely relevant for the special issue to which it was submitted. The challenges faced by Roma people (an important community in the EU by the figures of the last census) have considerably intensified during the pandemic. 

The introduction is well written and structured. I appreciate the five pillars structure namely:  New forms of family participation and incorporation of family knowledge, Organization in communities of practice, Personalization of teaching, Continuity of school practice through digital media, Relocation of school practice. 

The results are based on a sound methodology describing the context, the participants and the analysis.

In the discussion section I might suggest a further emphasis on the most vulnerable groups within the Roma community: girls, young moms etc. I would be interesting to know if there were any specific coping mechanism for these subgroups.

Overall the article is well written and provides a good contribution to the scholarly literature. Practitioners in school management would find it helpful. 

Author Response

Dear reviewer, 

thanks for your careful review. The changes proposed clearly help improving our manuscript. Bellow I am enlisting the changes made to our manuscript, following your and other reviewer´s suggestions:

Abstract has been revised to ensure it includes purpose, method, key findings and implications.

The introduction has been reorganized into three sub-sections, with the purpose of clarifying the theoretical referents in three areas. The first two subsections have to do with the state of the art and the last one with theoretical development based on team practice. Accessible sources/references in English have been included, counterbalancing the excessive bibliography in Spanish. Self-references have also been reduced to those strictly necessary (when we refer to the background of this line of research).

The method has been revised to clarify the relationship of the study with its context, the analysis procedure, and the referents on which it is based.

In the results, the name of the schools has been anonymized, although the reference to the cities in which they are located is maintained, as it may be relevant for researchers for whom the differential local elements of the Spanish educational system are important, managed in a regional level.

The discussion has been expanded to connect more explicitly with the theoretical elements presented in the introduction, developing especially the issues related to community resilience.

The conclusions have been simplified with a new wording, adding more explicitly the limitations of the study.

Finally, some typographical errors have been corrected and an external English revision has been carried out.

Change control has been activated so that there is a complete record, except for the change of position of two paragraphs to prevent them from appearing as new contributions.

Bests

Reviewer 4 Report

Dear Author, Thank you. I have read your paper which needs a significant amount of revision before it can reach to a publishable standard. I note my comments for the improvement of each section.

Abstract: While the Sustainability doesn’t want a structured abstract, however an unstructured abstract should also cover purpose, method, key findings and implications. You don’t have all these components in the abstract. Please rewrite the abstract by ensuing that all elements are covered.

Introduction is very long and it has not developed any sub-titled. With your current writing, it is very difficult to spotlight the research, problem, gap, scope, aim and objectives along with research question. It is hard to follow. I would also suggest you to develop a literature review section. You need a visit relevant article in the field and need to make a best use of them. Here are some examples, please ensure a best collection and make a best use of them:

# A COVID-19 Pandemic Sustainable Educational Innovation Management Proposal Framework- https://doi.org/10.3390/su13116391

# Online technology: Sustainable higher education or diploma disease for emerging society during emergency—comparison between pre and during COVID-19- DOI: 10.1016/j.techfore.2021.121034             

# Sustainability in Higher Education during the COVID-19 Pandemic: A Systematic Review- https://doi.org/10.3390/su14031879

# Access, attendance and performance in urban K8 education during pre- and post-COVID-19 restrictions in Bangladesh: comparison of students in slums, tin-sheds and flats- DOI: 10.1080/03004279.2022.2109183

You have not justified your method. This needs a to be justified. The writing is more descriptive that needs to be précised. Limitation is not acknowledged. It is very hard to justified which voice that you considered for data and which you didn’t and why. Please explain this.

In presenting findings, you need to response your research question by bringing the theme. You should not just report the findings without answering a particular point. Also need to add the discussion with your standpoint.

The implication part of this research rather weak, you need to report both theoretical and practical implication in order to arrive a conclusion.

The papers need English editing.

Please revise the before and I shall be happy to read the revision                

Author Response

Dear reviewer, 

thanks for your careful review. The changes proposed clearly help improving our manuscript. Bellow I am enlisting the changes made to our manuscript, following yours and the rest of the reviewer´s suggestions:

Abstract has been revised to ensure it includes purpose, method, key findings and implications.

The introduction has been reorganized into three sub-sections, with the purpose of clarifying the theoretical referents in three areas. The first two subsections have to do with the state of the art and the last one with theoretical development based on team practice. Accessible sources/references in English have been included, counterbalancing the excessive bibliography in Spanish. Self-references have also been reduced to those strictly necessary (when we refer to the background of this line of research).

The method has been revised to clarify the relationship of the study with its context, the analysis procedure, and the referents on which it is based.

In the results, the name of the schools has been anonymized, although the reference to the cities in which they are located is maintained, as it may be relevant for researchers for whom the differential local elements of the Spanish educational system are important, managed in a regional level.

The discussion has been expanded to connect more explicitly with the theoretical elements presented in the introduction, developing especially the issues related to community resilience.

The conclusions have been simplified with a new wording, adding more explicitly the limitations of the study.

Finally, some typographical errors have been corrected and an external English revision has been carried out.

Change control has been activated so that there is a complete record, except for the change of position of two paragraphs to prevent them from appearing as new contributions.

Bests

Round 2

Reviewer 1 Report

Thank you for refining the manuscript based on the reviewers' comments. The manuscript has developed a lot. My only comment is related to the titles: please check the full-stops (e.g., 1.1). Good luck with your further research. 

Author Response

Dear reviewers,

Thanks for your comments. Below, changes made in our manuscript following your last suggestions.

Tittle has been changed so it reflects paper content more accurately (lines 2 and 3).

Information about the sampling method and data analysis method have been included in the abstract (lines 14 to 17).

Section 1.1. has been revised in order to include a full stop between line 49 and 50 (new paragraph starting in “In Spain”)

Research question (line 198) and problem (line 195) explicitly stated in introduction

Practical and theoretical implications were already mentioned in the previous manuscript, in lines 799 to 803. For length reasons, we have not extended these implications further, but we have included a source that relates to the theoretical concepts referred for further exploration in line 803.

Reviewer 4 Report

Improvement is not up to the mark 

Author Response

(The authors gave the same response as above.)
